# Do a Learner's Background Languages Change with Increasing Exposure to L3? Comparing the Multilingual Phonological Development of Adolescents and Adults

Christina Nelson 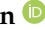

English Department, University of Münster, 48143 Münster, Germany; c.nelson@uni-muenster.de

**Abstract:** The present study longitudinally explores regressive phonological cross-linguistic influence (CLI) in seven adolescents (aged 12–13) and seven adults (aged 21–39) by examining voice-onset time (VOT) of /p,t,k/ in their first, second, and third language (L1, L2, and L3, respectively). All participants had the same language combination (L1 German, L2 English, L3 Polish) and were recorded completing a range of production tasks in all three languages four times over the course of the first year of L3 learning. The scope of previous research on phonological CLI is thus broadened in two ways: (1) by tracing the development of *all* languages upon the arrival of a new language in a multilingual's system longitudinally, and (2) by investigating CLI patterns in two age groups when input and learning environment are comparable. Previous L2 age studies have mostly only made retrospective assumptions about (target) language development, so that longitudinal data, including the entire language repertoire of multilingual speakers, are needed to substantiate claims made in that regard. For the adolescent group, significant changes to both their L1 and L2 over time were found, while the adults' background languages remained relatively stable on the group level. However, for both groups, much individual variation was uncovered.

**Keywords:** regressive CLI; multilingual language development; L3 learners; adults; adolescents; phonology; VOT; phonological permeability hypothesis; instructed learning context

## 1. Introduction

Arguably, from time to time every person who speaks more than one language experiences phenomena such as lexical access difficulties, using another language's syntax, or applying phonological rules from another language. Such striking instances of cross-linguistic influence (CLI), the interaction of a speaker's languages, are only the tip of the iceberg. As a plethora of studies over the last decades have demonstrated, languages constantly interact and influence each other in the multilingual mind (Cenoz et al. 2001). Accordingly, rather than rigidly separated entities, a speaker's languages are presumed to exist within a complex and interconnected system (De Bot 2012).

The majority of previous studies has mainly focused on the progressive kind of CLI, i.e., from an earlier learned language onto a subsequent one (see, e.g., De Angelis 2007, p. 20; Wunder 2011, p. 107). However, CLI also occurs the other way around, namely from a later learned language onto an earlier one. In fact, we know from a solid body of attrition research that even a speaker's L1 is not as stable as once presumed. Language attrition is a well-documented phenomenon in the context of speakers who spend extended periods abroad and, as a result, experience changes or problems with their native language. Köpke and Schmid (2004, p. 5) define attrition much more generally as a "non-pathological decrease in a language that had previously been acquired by an individual". The vast majority of language attrition studies focus on immigrant contexts and only take L1 attrition into account, but several L2 phonology studies conducted in this line of thinking have suggested that the development of a new language system, even in low proficiency instructed learners,

will impact the L1 phonology to some degree (cf. Schmid and Hopp 2014, p. 371). These mostly temporary and subtle changes of a learner's pronunciation in their L1 have been termed "gestural drift" (Sancier and Fowler 1997) or "phonetic drift" (Chang 2012), and there seems to be a growing interest in this phenomenon.

Naturally, there is even *more* potential for cross-linguistic interaction of this kind in the case of third language (henceforth L3) learners, who have at least two already existing language systems in which changes could be triggered by the onset of learning a new language. With more (three plus) languages in the mix, much more complex and diverse patterns of mutual interaction are conceivable for multilingual learners in comparison to those who are only learning their first non-native language. An example for this is combined CLI, which is "when two or more languages interact with one another and concur in influencing the target language, or whenever one language influences another, and the already-influenced language in turn influences another language in the process of being acquired" ((De Angelis 2007, p. 49); for examples of this kind of CLI see, e.g., Pyun (2005) and Blank and Zimmer (2009)).

One of the important questions with regard to regressive influence in multilingual learners is what factors condition this form of CLI and which of a learner's languages are more susceptible to it than others. In the few L3 studies conducted in this line of inquiry, the effect of language status is discussed frequently. Although their findings are somewhat mixed and exhibit much individual variation, there seems to be a tendency for more L3→L2 than L3→L1 influence (see Section 1.3). It should be noted, however, that these studies are rather exploratory and thus naturally limited in the number and population of speakers, learning contexts, and features under investigation.

For instance, very little is known about child and adolescent speakers at this point, despite the fact that age as a variable is particularly relevant when it comes to regressive CLI. In this connection, hypotheses about language status have been raised, stating that a person's L1 may be more stable and thus less malleable or vulnerable than even a native-like L2 due to maturational constraints, i.e., as a result of age of learning (see Section 1.3 for a discussion of Cabrelli's (2016) Phonological Permeability Hypothesis). Age in this context is usually conceptualized as the binary differentiation of pre- versus post-pubescent, and its effect on CLI has only ever been assessed 'retrospectively' in adult speakers, for example by comparing mirror-image groups of early and late adult bilinguals at some point during the early stages of L3 learning (e.g., Cabrelli 2016; Cabrelli and Rothman 2010). While such investigations are insightful with regard to other research questions (e.g., those of maturational constraints of language learning), they are more product-oriented in nature and therefore do not shed much light on how multilinguals of different age groups develop in their languages over time. Moreover, due to the common onset of additional language learning at around 11–13 years in the educational systems of many countries, there is a large number of L3 learners in instructed learning contexts that evade the dichotomy of pre- and post-pubescent learners, as they virtually fall in the middle of the two. This group has, to date, been majorly underrepresented in phonological L3 research in general (but see Kopečková 2016; Llama and López-Morelos 2016; Nelson 2020; Nelson et al. 2021; Kopečková et al. 2021).

Adolescent L3 learners are different from the usual adult population of speakers featured in L2 or L3 studies in many respects and can thus be expected to exhibit other CLI patterns. For one, they normally still receive formal instruction in their L2 alongside the new language and can be located at lower or intermediate L2 proficiency levels, whereas adult multilinguals typically have higher L2 proficiency levels and do not receive instruction in that language anymore. It is therefore reasonable to assume that the adults' L2 system is *relatively* more stable than the adolescents' system. Some evidence for this claim for this very group of learners is provided by Nelson (2020)[1], who showed that the adults' perceptual accuracy of the non-native /v-w/ contrast in their L2 and L3 did not nearly vary or fluctuate as much as for the adolescents. This is not to say, however, that the adults' L2 system would be expected to have reached a stage of a complete standstill. Changes over

time have also been attested for highly proficient adult speakers (see, e.g., Beckmann 2012), for instance triggered by external events or conditions as a shift in language environment because of travelling or moving or starting to learn another language. Still, it can be argued that such events have an even stronger impact on an L2 that is still in the process of *actively* being learned, as usually is the case for the younger learners. Secondly, younger learners may also differ with regard to the status of their L1. As phonological perception studies in related fields, such as audiology, have demonstrated, even adolescents do not quite perform the same as adults, especially when tested under challenging conditions like reverberation and noise (Bent 2015; Hazan and Barrett 2000; Johnson 2000). This suggests that a person's L1 continues to develop throughout childhood and into puberty, which again possibly makes it more malleable and thus more susceptible to influences from other languages a speaker encounters.

However, one aspect that applies to all learners, regardless of age, is that their language system is likely to self-organize as a result of learning a new language, yielding unique learning trajectories for every speaker. An insightful perspective through which this can be studied and understood is offered by the Complex Dynamic Systems Theory of language development (CDST). Rather than conceptualizing language learning in a product-oriented and often linear way, it understands it as a dynamic process (Beckner et al. 2009; Larsen-Freeman 2015). Therefore, the term 'language acquisition' is replaced by 'language development' in that framework, as 'acquisition' implies that there would be an end point to learning at which the learner's language reaches stability: "By using 'development' rather than 'acquisition', we want to make it clear that linguistic skills can grow and decline, and that accordingly, language acquisition and language attrition are equally relevant outcomes of developmental processes" (De Bot and Larsen-Freeman 2011, p. 6). According to CDST, all of a speaker's languages are assumed to exist in an interconnected, adaptive system that perpetually interacts with its environment and reorganizes itself. While such a system may also undergo periods of stability, which are also called attractor states, "there is no stasis, and the system has the potential to undergo radical change at any time" (Larsen-Freeman 2017, p. 16). Within this framework, regressive CLI would not only be deemed possible, it is almost to be expected that starting to learn a new language would trigger some changes in the already existing language subsystems. It is thus not surprising that there are a number of publications that make the explicit link between CDST and L1 attrition research (see, e.g., Herdina and Jessner 2002; De Bot 2004; Schmid et al. 2012; Leeuw et al. 2013; Köpke 2017; Opitz 2017). However, due to the complexity of the system's interactions with its numerous variables and its sensitive dependence on initial conditions (De Bot and Larsen-Freeman 2011, p. 10), it is difficult if not impossible to predict in what ways regressive CLI would surface for a particular learner. Understanding language learning processes and CLI through this lens therefore requires zooming into an individual's learning trajectory in several subsystems as opposed to just analyzing a group's performance as a whole in a target language. As Larsen-Freeman (2017, p. 27) puts it, "[e]ach developmental trajectory is unique. Therefore, we can make claims at the level of group, but we cannot assume that they apply to individuals".

The present study aims to investigate regressive phonological CLI in seven adolescents (aged 12–13) and seven adults (aged 21–39) with the same language combination of L1 German, L2 English, L3 Polish. They were recorded completing a range of production tasks (delayed repetition, picture-naming, and storytelling) in their L2 and L3 four times over the course of the first year of instructed L3 learning (after one, three, five, and ten months of learning), and additionally in their L1 at the beginning and end of the year. The feature reported on in this paper is voice-onset time (VOT), which typically differs for the three languages (see Section 1.4). It is hypothesized that: (1) for both groups, the L2 will be more vulnerable to L3 influence than the L1 due to relative differences in stability, and (2) that if changes of the L1 do occur, they will be more visible for the adolescents than the adults. Beyond such group comparisons, one of the study's main foci lies on the exploration of individual trajectories in all three languages. It thus hopes to make a contribution to a

comprehensive model of multilingual language development in the future by exploring two lesser-studied aspects of it, i.e., emergent multilinguals of different ages and regressive CLI. Previous studies relevant to these topics are discussed next, starting with regressive CLI in L2 and L3 studies.

*1.1. Regressive CLI in L2 Studies*

There is a considerable number of studies in the L2 context that explore the circumstances and conditions under which regressive phonological and phonetic CLI from a person's L2 to their L1 takes place (see, e.g., Alves et al. 2019; Barlow 2014; Chang 2012, 2013, 2019; Cook 2003; Flege and Eefting 1987a; Jiang 2010; Kartushina et al. 2016a; Lev-Ari and Peperkamp 2013; Lord 2008; Major 1992; Marx 2002; Mennen 2004; Sancier and Fowler 1997; Schereschewsky et al. 2018; Stoehr et al. 2017; Tobin et al. 2017; Ulbrich and Ordin 2014; Waniek-Klimczak 2011). Most of these studies performed perceptual or acoustic analyses to deliver a quantitative assessment of L2 influence on L1 production, for instance, by comparing sequential bilinguals to a monolingual group or previously reported literature values (for a comprehensive literature review see Kartushina et al. 2016b). They have shown that changes to a speaker's L1 can manifest in seemingly opposite ways. The most straightforward option is the finding that realizations of segments in the L1 'drift' towards their L2 counterparts (e.g., Barlow (2014) for use of dark /l/; Chang (2012, 2013) for VOT and vowels; Flege (1987) for VOT; Kartushina et al. (2016a) for vowels; Lev-Ari and Peperkamp (2013) for VOT; Major (1992) for VOT; Sancier and Fowler (1997) for VOT and foreign accent ratings). On the other hand, some studies (e.g., Flege and Eefting (1987a, 1987b) for VOT; Guion (2003) for vowels) also found that L2 speakers' L1 realizations differed significantly from control groups or literature values in a way that did not directly point to an underlying L2 influence, but instead could be interpreted as a deflection of L1 categories *away* from similar L2 sounds.

While it is not entirely clear why speakers develop in one way or the other, Kartushina et al. (2016b, p. 170) hypothesize that the direction of learners' development is dependent on the learning and speaking context:

> It is possible that in those proficient L2 speakers who are immersed in the L2 environment, the L2 serves as a magnet that attracts L1 phonetic production (e.g., as in the study by Major 1992), whereas that in individuals who are not immersed in the L2 environment (e.g., as in the study by Flege and Eefting 1987a), L1 phonetic production is deflected away from L2 sounds as a means of keeping the two phonetic repertoires distinct. Immersion is very often associated with L2 dominance, and with more code mixing (i.e., language mixing), factors which together favor the assimilation, in production, of the L1 toward the L2 [ . . . ].

Although more systematic investigation is needed to confirm this hypothesis, it is beyond doubt that learning context and amount of use generally play an important role, as they likely modulate the extent to which changes in the L1 occur. For instance, sustained immersion in an L2 environment often entails a decrease in L1 use as well as an increase in L2 use and proficiency, all of which are factors that are known to boost attrition effects (see, e.g., Flege 1987; Leeuw et al. 2010; Schmid and Dusseldorp 2010). On the other hand, it is less certain what factors have to be present in order for regressive CLI to occur in non-immersive or typical classroom learning contexts where exposure is more restricted, as is the case for the learners in the present study. Still, research conducted in instructed learning contexts and L1-dominant speaking environments (e.g., Flege and Eefting 1987a, 1987b; Kartushina et al. 2016a; Lord 2008; Alves et al. 2019; Herd et al. 2015; Kupske 2016; Schereschewsky et al. 2018; Waniek-Klimczak 2011) has uncovered a number of interesting observations and tendencies.

First of all, there is evidence that regressive CLI in the form of an L1 phonetic drift can take place after only minimal input for perceptually similar sounds. Kartushina et al. (2016a) investigated the mutual influence between native and non-native vowel production in 20 L1 French speakers. They were recorded before and after three articulatory

training sessions (amounting to a total of three hours) with two non-native vowel sounds of languages they had never had any exposure to prior to the study, Danish /ɔ/ and Russian /ɨ/. For the L1 French /ø/ vowel, regressive CLI was detected in the form of a drift of F1/F2 measures towards those of the similar Russian vowel /ɨ/. This finding is especially noteworthy considering that these two vowels are not as perceptually or acoustically similar as, for example, Danish /ɔ/ and French /o/. Therefore, the authors conclude that non-native sounds have to be perceptually close to, but not prototypical of, the native one for phonetic drift to occur. The fact that changes to the speakers' L1 systems were detected after minimal exposure of three hours to an entirely novel sound highlights the adaptability and dynamicity of L1 categories even outside of immersive or immigrant contexts. It is unclear, however, how temporary this training-induced effect was and to what extent their hypothesis regarding which sounds are prone to regressive CLI can be generalized to other structures of speech. More conventional studies with 'ordinary L2 learners' living in an L1-dominant environment may be informative here.

Lord (2008) and Herd et al. (2015), shedding light on the influence of L2 proficiency, both found L2-like VOTs for some stops in advanced speakers of L1 English and L2 Spanish. The former only tested advanced speakers in the first place (and only found statistically significant differences for the stop /k/, not for /p/ or /t/), while the latter compared groups of different proficiency levels and reports the "most robust evidence of phonetic drift" for the near-native group (Herd et al. 2015, p. 7). However, it is also noteworthy that group differences in VOTs in Herd et al.'s (2015) study were already visible between beginner (first semester of learning Spanish) and intermediate learners (third semester of learning Spanish). This finding indicates that even in L1-dominant, instructed learning environments, regressive CLI in the form of phonetic drifting can take place within the first one or two years of learning.

Yet another interesting hypothesis on the potential workings of regressive CLI is brought up by Alves et al. (2019). They conducted a similar study, testing ten speakers from Argentina with a reverse language profile to Lord's (2008) and Herd et al.'s (2015) participants—with L1 Spanish and advanced L2 English. When they compared their VOTs to a group of ten monolingual L1 Spanish speakers, they found that the bilinguals produced significantly higher VOTs for /k/ and /p/. Indeed, the velar stop appears to be the first place of articulation to exhibit signs of regressive CLI for L1 Spanish as well as L1 Brazilian Portuguese speakers (Kupske 2016; Schereschewsky et al. 2018). Apparently, this effect may also work the other way around, with L1 English /k/ being more vulnerable to change than the other voiceless plosives, as demonstrated in Lord (2008). This suggests that the occurrence of regressive CLI may interact with and mirror patterns of articulatory contextual factors.

Waniek-Klimczak (2011) discusses how the two may be intertwined in her study investigating VOT of two voiceless plosives (/p,k/ followed by /i,a/) of 20 L1 Polish speakers. Half of them had minimal exposure to English (Group 1), whereas the other half consisted of proficient L2 English speakers who use English in their everyday life in Poland (Group 2). She employed a word-reading task as well as a less-controlled, more emphatic dialogue reading, the latter of which she predicted to be more conducive to (English-like) aspiration as a result of a stylistic effect of emphasis, "based on implicational, 'ecological' universals" (Waniek-Klimczak 2011, p. 6). The results show a clear effect of L2 experience, with consistently lengthened (and thus English-like) VOTs in all contexts for Group 2 in comparison to Group 1. However, concerning task-related differences and the effect of style, contrary to prior predictions, the more carefully produced word list tokens elicited slightly longer VOT values than the dialogue reading. This was especially the case when the plosive was followed by /i/, as would be expected according to phonetic universal factors. However, the more proficient group also tended to aspirate in non-high vowel contexts (i.e., in an L2-like manner). Another intriguing observation emerges when Waniek-Klimczak compares her data to those of Keating et al. (1981), who tested speakers with the same profile: The mean values for both plosives are higher for both groups in her study

(although more so for the group with more L2 English experience), which she tentatively interprets as evidence for a general language change in progress among young, highly educated Poles, potentially introducing voiceless aspirated stops as one of the phonetic categories in Polish. While more research is necessary to confirm this tendency, this study aptly illustrates some of the opportunities and challenges inherent to the investigation of regressive CLI. Its findings reiterate that language change exists on two intertwined levels, that of the individual speaker and that of a larger group that shares certain extra-linguistic characteristics (for an interesting juxtaposition of the two, see, e.g., Baxter and Croft 2016). As a result of globalization and increasingly multilingual societies (Aronin 2018, p. 27; Hammarberg and Williams 1993, p. 60), processes of language change may emerge on both levels. For research on regressive CLI, the challenge remains to discern and categorize occurrences of changes to a speaker's language.

To summarize, there is a considerable body of research attesting to L2-related changes to a speaker's L1 in immersive as well as instructed L2 learning environments. These changes can manifest in manifold ways, e.g., as a drift of L1 structures towards or a deflection away from similar L2 categories. Amount of language use, target language input, and proficiency are all among the potential factors that seem to determine the magnitude of such L2 effects. While these studies have laid important groundwork for the investigation of regressive CLI, a whole new set of questions arises in the context of L3 learning.

### 1.2. Regressive CLI in L3 Studies

Apart from some unexpected discoveries of regressive phonological CLI in L3 studies that were designed to analyze forward transfer (e.g., Llama and Cardoso 2018; Wrembel 2015), there have been seven studies over the previous decade that investigate different types of regressive CLI in adult learners (see Table 1). All of them measured either vowels or VOT in speakers who were permanently living in the L2/L3 environment or had just spent an extended amount of time there (except for Aoki and Nishihara (2013) and Sypiańska (2016)). Their L2 and L3 proficiency varied somewhat but, with a few exceptions, can mainly be located at an (upper-)intermediate level. The studies can be categorized into three groups according to the exact type of CLI that was the center of interest:

1.  Focus on L3-induced changes to the speakers' L2 only (Aoki and Nishihara 2013);
2.  Investigation of L3 as a source of regressive CLI for the other languages, the main question being whether it (rather) influences L1 or L2 (Beckmann 2012; Cabrelli 2016; Cabrelli and Rothman 2010);
3.  Investigation of all possible kinds of regressive CLI, so including L3 influence onto L1 and L2 as well as L2 onto L1 (Liu 2016; Sypiańska 2016, 2017).

**Table 1.** Studies investigating regressive CLI in L3 speakers (in chronological order).

| Publication, Type of Investigated CLI | Speakers | Feature(s) | Findings and Effect Sizes |
|---|---|---|---|
| Cabrelli and Rothman (2010), L3→L1/L2 | 1 simultaneous, 1 successive bilingual with English/Spanish, both learning L3 Brazilian Portuguese | Nasalization, spirantization, vowel neutralization, coda treatment (in S and BP) | L3→L2 quick and pervasive in successive bilingual; no such influence in simultaneous bilingual. Descriptive pilot study, no effect sizes reported. |
| Beckmann (2012), L3→L1/L2 | 7 frequent L3 users, 7 less frequent L3 users (all with L1 German, L2 English) | VOT in all languages | L3→L2 influence in frequent L3 users only (significant group difference, with L3-like values for the frequent L3 users). No evidence for L3→L1 influence in either group. No effect sizes reported. Much individual variation. |

**Table 1.** *Cont.*

| Publication, Type of Investigated CLI | Speakers | Feature(s) | Findings and Effect Sizes |
|---|---|---|---|
| Aoki and Nishihara (2013), L3→L2 | 6 speakers of L1 Japanese, L2 English, L3 Chinese (plus 6 non-native control speakers without L3 Chinese and 2 native control speakers) | VOT in L2 English | Facilitative L3→L2 influence (despite relatively low L3 proficiency), leading to better performance than controls without L3. A medium effect size of $\eta^2 = 0.098$ was reported for the significant group effect. |
| Cabrelli (2016), L3→L1/L2 | 15 speakers of L1 English, L2 Spanish, 8 speakers of L1 Spanish, L2 English (to isolate factor AoA), everyone with L3 Brazilian Portuguese | Vowel reduction (perception/production) for /e/ (F1–F0 and F2–F1) and /o/ (F1–0, F2–F1, and duration) | *Production:* Spanish as L2 more susceptible to influence from L3 BP than Spanish as L1 on some measures: significant group differences with small effect sizes reported for /o/ (F1–F0) and /e/ (F2–F1), but not for /e/ (F1–F0) or /o/ (F2–F1 and duration). Much individual variation. *Perception:* No effect. |
| Liu (2016), L3→L1/L2, L2→L1 | 2 groups of L1 Chinese, L2 English speakers (residing in Spain): <br>• 10 with L3 Spanish (B2/C1 level) <br>• 10 without any L3 instruction | VOT of bilabial stops (perception/production) | *Production:* Evidence for L3→L2 effect (however, albeit significant, group differences were numerically small). *Perception:* regr. CLI from L2/L3→L1 (both groups had different L1 perceptual boundaries than monolingual control group), but not L3→L2 (no difference between the experimental groups). No effect sizes reported. |
| Sypiańska (2016), L2/L3→L1, L3→L2 | 3 groups of L1 Polish speakers: <br>• 11 with L2 English, L3 German <br>• 11 with L2 English, L3 Spanish <br>• 11 with L2 German, L3 English | Vowels | No evidence for L3→L2 effect (as no differences found between Group 1 and 2). L1 values differed from baseline in all groups: L2→L1 effect in Group 1 and 2, L2/L3→ L1 effect in Group 3. No effect sizes reported. |
| Sypiańska (2017), L2/L3→L1, L3→L2 | 2 groups of Polish immigrants in Denmark: <br>• 10 with L1 Polish, L2 Danish <br>• 20 with L1 Polish, L2 Danish, L3 English | Vowels and VOT | L2→L1 effect for both features and groups. In the multilingual group, L3→L1 influence was less visible than L3→L2 and L2→L1. No effect sizes reported. |

Aoki and Nishihara (2013), in the only paper that exclusively focused on lateral-regressive CLI (L3→L2), report that six speakers of L1 Japanese, L2 English, L3 Chinese outperformed a control group without L3 Chinese in terms of target-likeness of L2 VOTs. The authors interpreted this as facilitative regressive influence from L3 Chinese. The three studies subsumed under (2), which considered L3-induced changes to a speaker's L1 and L2, all provide evidence that the L3 is more likely to exert influence on another non-native language (in this case, the speaker's L2) than the native one, despite the fact that all participants used their L2 frequently and at an advanced level. However, much individual variation was attested as well, which points to the involvement of factors other than language status.

The findings of the studies in category (3) are more mixed than those in (2). Here, the authors included all possible types of regressive CLI, i.e., with L3 as a potential source, L2 as

both potential source and recipient (of L3 influence), and L1 as a potential recipient (of either L2 or L3 influence). The broader scope of interactions naturally makes for more complex results. Liu (2016), the only study besides Cabrelli (2016) that tested both production and perception, found some potential evidence for an L3→L2 effect in the form of significant differences between L2 VOTs of the two groups of Chinese L2 English learners with and without L3 Spanish instruction (but both residing in Spain as international students). The groups did not differ in their perception of VOT boundaries in L2, though, ruling out L3 influence. However, compared to a monolingual Mandarin control group, both the bi- and the trilingual group had lower perceptual boundaries in their L1. This may point to regressive CLI, although it is not entirely clear from the data whether the L2, the L3, or a combination of both may be responsible for that shift in boundaries.

Similarly, Sypiańska (2016) reports significantly different L1 vowel measurements for all three groups of trilinguals in her study (all with L1 Polish residing in Poland, but differing in their L2 and L3) in comparison to a monolingual control group. Based on the groups' language combinations, this is interpreted as L2→L1 effect in two of the groups, while the source language(s) cannot be clearly identified in the third group (either L2, L3, or both combined). Interestingly, in contrast to the studies in category (2), no L3→L2 influence was detected (which would have manifested itself in group differences between the two groups for whom only the L3 differed). In another study, this time with L1 Polish immigrants in Denmark (i.e., with L2 Danish) either with or without L3 English, Sypiańska (2017) once again found evidence for L2→L1 influence. This discovery is not surprising considering that this population would be a typical target group of traditional attrition studies for whom this is a well-documented phenomenon. In addition, altogether echoing previous studies' findings, for the multilingual group, she reports more instances of regressive CLI in the form of L3→L2 and L2→L1 influence than L3→L1. These results denote a mixture of several factors at play, such as the dominant language environment of the speakers (explaining attrition effects in the form of L2→L1 influence), as well as (perceived) similarity or language status (accounting for the finding of L3→L2 rather than L3→L1 influence). The speakers' two non-native languages L2 Danish and L3 English are phonologically closer and, due to their shared non-native status, perhaps also cognitively more similar to each other than to their L1 Polish. Although these conditions seem to be at odds with the clear finding of L2→L1 influence at first glance, it is conceivable that language environment and use possibly have such a significant impact that this interaction takes place even in the absence of other boosting factors.

To summarize, findings concerning regressive CLI in multilingual speakers, although somewhat mixed, indicate a trend of more L3→L2 and L2→L1 than L3→L1 influence. A fair deal of individual variation is also reported in most studies. From a CDST perspective this is not surprising, as language development is considered to be a complex and self-organizing process. This warrants a closer look at individual data (as opposed to just group results) and developmental trajectories.

### 1.3. The Phonological Permeability Hypothesis

From the preceding discussion of previous research in the field, it transpired that certain factors condition regressive CLI in multilingual speakers, including language status, language environment and use, and (perceived) proximity. Among these, language status or, by implication, age of acquisition, appear to be recurrent, strong predictors. Accordingly, Cabrelli and Rothman (2010) offer a hypothesis concerning phonological CLI in multilingual learners. The Phonological Permeability Hypothesis (PPH) holds that "L1 and L2 systems are fundamentally different, and that this difference is maturationally conditioned" (Cabrelli 2016, p. 2). However, this difference does not refer to a maturationally constrained access to language universals in second or third language acquisition, but to a difference in stability[2] between a learner's L1 and L2 systems. Hence, according to the PPH, "even an ostensibly native-like L2 is more vulnerable to L3 influence than an L1" (Cabrelli 2016, p. 699). This is not to say that the L1 could not potentially be affected as well; it is merely

proposed that changes to the L2 would be more pervasive and occur faster in comparison due to its greater permeability.

Cabrelli (2016) examined the acquisition of L3 Brazilian Portuguese (BP) by two types of sequential bilinguals in a mirror-image design (fifteen L1 English/L2 Spanish speakers, eight L1 Spanish/L2 English speakers). With this design, she was able to test the level of BP influence on L1/L2 Spanish perception and production, isolating age of acquisition as a factor. The results of the Spanish production tasks indicate a difference between the L2 Spanish group (visible influence of BP in the height of back vowels) and the L1 Spanish group (height remained the same). This finding is in line with some other studies discussed above, which either reported more L3→L2 than L3→L1 influence, or no L3→L1 influence at all (see, e.g., Beckmann (2012); Liu (2016) for production only). However, no evidence supporting the PPH was found in perception, which she tested as well. Therefore, Cabrelli (2016, p. 14) states that the study

> does not support the PPH in its original form since there is no evidence of changes to mental representation. Instead, the outcome indicates that the addition of a novel phonological system can affect aspects of speech production to a larger degree in late-acquired systems, at least when the languages under observation are typologically related.

Naturally, further testing is needed to substantiate this claim, e.g., including other language pairings, as the PPH proponents encourage others to do (Cabrelli and Rothman 2010, p. 293). In the present study, only the learners' L1 and L2 are typologically related (and arguably not to the same extent as Spanish and BP), so it remains to be seen how the languages interact in the absence of such straightforward and strong cues. Furthermore, while the L3 is not from the same language family as the L1 or L2, based on a concept of property-driven phonological typology as suggested by Hyman (2014), it can be argued in this specific case that L1 and L3 bear a greater phonological resemblance than L2 and L3 (see also Nelson et al. 2021, pp. 3–4). However, their similarity is likely not powerful enough to override the effect of language status, if that was possible.

In the initial proposal of the PPH, Cabrelli and Rothman (2010, pp. 282–83) make some methodological recommendations for future studies intending to test the hypothesis. Some of these recommendations are reflected in the present study, while others are not. In what follows, the most relevant ones are discussed. Although the PPH may not offer firm predictions for all aspects of this study as it is not geared towards this specific context, sensible prognoses can be deduced on its basis in some cases:

- *Testing of two groups with the same language repertoire but consisting of simultaneous and (post-puberty) sequential bilinguals.* This is advised in order to isolate age of acquisition as a factor, which can also be achieved with alternative methods (see, e.g., Cabrelli's (2016) mirror-image design). Importantly, the PPH concentrates on differences in mental constitutions between pre- and post-pubescent learners (Cabrelli and Rothman 2010, p. 278), now manifested in diverging CLI patterns. While the factor age is of interest in the present study as well, a different angle is attempted. First of all, the focus here lies on the multilingual language development and CLI patterns of different age groups observed in 'real-time' as the objects of interest themselves, rather than aiming to settle the long-standing debate of maturational constraints and resulting mental constitutions. Secondly, this study extends beyond the dichotomy of pre- vs. post-pubescent learners by including young adolescent learners (aged 12–13) who fall precisely in the middle of the two. Despite these obvious differences between the PPH's original premises and the setup of the present study, there are still some predictions regarding the effects of age of L2/L3 learning and language status that can be deduced from the PPH. Generally, based on language status, more L3→L2 than L3→L1 influence can be expected for both groups. In addition, since the PPH reasons that greater L2 permeability might be due to differences in stability between L1 and L2 systems, it can be argued that the younger speakers' L1 might be relatively more susceptible to regressive CLI from any language, for that matter, than the adults' L1.



As mentioned above, additional support for this reasoning can be found in studies demonstrating that a person's L1 continues to develop throughout childhood and into puberty (see, e.g., Bent 2015; Hazan and Barrett 2000; Johnson 2000).

- *Testing speakers who are highly proficient in their L2.* The ideal group of learners initially envisioned for testing the PPH would partly consist of native-like L2 speakers (Cabrelli and Rothman 2010, p. 281). While recruiting such a group of learners would be highly informative regarding the aim of uncovering differences between L1 and L2 systems that, on the surface, appear indistinguishable, this seems an ambitious endeavor: Perfectly native-like L2 speakers are difficult to find in the first place, and in addition, they would also have to be beginning learners of the same L3. In her follow-up study testing the PPH with a larger pool of participants, Cabrelli (2016) did not limit her inclusion criteria so drastically. However, she carefully tested that the learners had acquired the features under investigation in their L2 and L3, and also employed a general L2 proficiency test and foreign accent ratings. Again, due to the different focus of the present study, this was not at all considered as a potential inclusion criterion. Yet, another intriguing aspect in relation to L2 proficiency arises from the two groups included here. There is a clear (age-related) contrast between the age groups in that regard, with higher L2 proficiency levels among the adults. The adolescents were still very much in the process of 'actively acquiring' the L2, receiving formal instruction alongside their L3 classes.

- *Collect longitudinal data.* In order to trace the development of interactions over time, both L2 and L3 should be tested multiple times. More specifically, Cabrelli and Rothman (2010, p. 282) propose monthly intervals over the course of the first year of exposure in a naturalistic setting. According to their predictions for simultaneous vs. sequential English/Spanish bilinguals acquiring Brazilian Portuguese, both groups would initially exhibit significant Spanish→BP transfer regardless of language status due to the two languages' typological proximity. This influence is predicted to diminish relatively quickly for the late bilinguals and more slowly for the early bilinguals (for the latter group, it is thought to persist considerably longer because of the native status of their Spanish system). While the group of late bilinguals is expected to begin experiencing some steadily increasing L3→L2 influence early, only very little regressive CLI is forecast for the early bilinguals. To date, there has not been a longitudinal investigation to confirm or falsify these predictions of an interaction trajectory. Generally, longitudinal studies are rare to begin with (and perhaps more so for phonological research), let alone testing the same cohort of participants twelve times over the course of one year, each time with a lengthy testing battery including two or more languages and, as is discussed below, a variety of different phenomena in both modalities. Therefore, although the proposed methodology would certainly yield interesting insights regarding cross-linguistic interactions and language development, its feasibility can be called into question. In the present study, a compromise was implemented, collecting data longitudinally in all languages in the first year of L3 learning, but only with four testing times.

- *Apply both auditory and acoustic analyses.* Extrapolating from the evidence brought forward by numerous studies on phonological L1 attrition, occurrences of regressive CLI may not manifest themselves in stark before/after contrasts. Instead, "changes in the learners' production could be subtle and imperceptible to even a trained ear" (Cabrelli and Rothman 2010, p. 289). Accordingly, it is advisable to supplement auditory analysis methods with acoustic ones, which may be better suited to capture small but systematic developments. For the purpose of the present study, VOT was chosen as the feature of interest as it may facilitate the detection of more subtle changes over time. The following section elaborates on VOT as a frequently studied aspect of phonological learning and its standing in the three languages featured in this study.

*1.4. The Feature under Investigation: Voice-Onset Time in German, English, and Polish*

Apart from vowels, VOT is considered the most commonly investigated segmental property of L2 speech (Edwards and Zampini 2008). This is also true of phonological CLI studies conducted in the multilingual context (Llama et al. 2010; Wunder 2011; Bandeira and Zimmer 2012; Beckmann 2012; Aoki and Nishihara 2013; Wrembel 2014, 2015; Gabriel et al. 2016; Liu 2016; Llama and López-Morelos 2016; Dittmers et al. 2017; Sypiańska 2017; Llama and Cardoso 2018; Amengual et al. 2019; Amengual 2021). What makes VOT so attractive to researchers, especially for those with cross-linguistic applications, is that it is a single acoustic measure which appears to correlate with the voicing contrasts in most languages (Cho and Ladefoged 1999). It has mainly been employed as phonetic measure of phonemic voicing in pre-vocalic stops.

According to previous studies, the languages spoken and learnt by the participants in the present study differ on their typically produced VOTs. Figure 1 visualizes the three languages' VOT continuums for /p,t,k/ as produced by monolingual speakers of the respective language. Generally, English and German can be classified as aspirating languages with long-lag VOTs, especially so in English. Polish, as a voicing language, typically has shorter ones. Although the VOTs produced by the learners in this study will not be directly compared to these monolingual literature values, they may be helpful in interpreting and discussing the data later on. However, rather than assessing the learners' target-likeness according to these norms, it is of interest in this study how *their* particular values change over time.

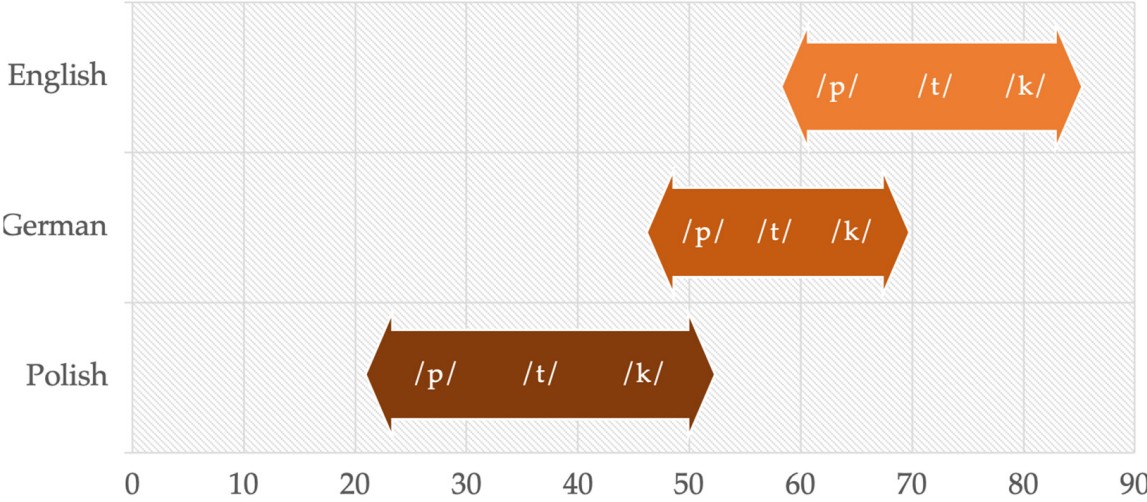

**Figure 1.** Visualization of VOT ranges in each language as reported for monolingual speakers in previous literature (Lisker and Abramson (1964) for English, Fischer-Jørgensen (1976) and Haag (1979) for German, Keating et al. (1981) for Polish).

*1.5. Research Questions and Predictions/Hypotheses*

The overarching research question the present study intends to explore is whether a multilingual learner's background languages, i.e., their L1 and L2, change over time with increasing exposure to L3. More specifically, the following research questions and hypotheses are posed:

- **Research Question 1 (RQ1).** *How do trajectories differ comparing the learners' L1 and L2?*

  **Hypothesis 1 (H1).** *The learners' L2 will be more vulnerable to L3 influence (in whatever way this influence surfaces) than their L1 (based on previous studies as well as the PPH (Cabrelli 2016)).*

- **Research Question 2 (RQ2**). How do trajectories differ comparing the adolescent and the adult learners?

**Hypothesis 2 (H2).** *The adults' languages will be relatively speaking more stable than the adolescents' languages. This should especially apply to the adolescents' L2, as they are still in the process of actively learning it, so that formed categories are more malleable and therefore more susceptible to change.*

- **Research Question 3 (RQ3).** How do trajectories differ for each learner individually?

It is expected that investigating the development of each learner (as opposed to just group values) will uncover much intra- and inter-individual variability. This research question is an exploratory one and, due to its open-ended nature, formulating a falsifiable hypothesis does not seem appropriate here. However, from previous CDST literature it can be hypothesized that the development between a learner's languages changes over time with increasing proficiency (Yu and Lowie 2019; Huang et al. 2020). It may for instance be competitive to begin with and turn out to be more supportive later on. What is more, extreme intra-individual variability at one point in time may signal an upcoming restructuring process of the system, resulting in major developmental changes (Evans and Larsen-Freeman 2020; van Dijk and van Geert 2007).

## 2. Materials and Methods

### 2.1. Longitudinal Design

Figure 2 visualizes the longitudinal data collection with four testing times. It began a month after the participants' start of L3 Polish lessons and stretched over the first year of learning. As more changes were expected at the earlier stages of L3 learning (as evidence of a reorganization of the language system due to the new input), the four testing times were selected with increasing time intervals between them instead of spacing them out evenly over the year. Such a longitudinal design is extremely informative when tackling research questions regarding learning trajectories, as every learner can act as their own control (Cabrelli 2013, p. 103). The assessment and exploration of group data can then be supplemented by consulting individual trajectories as well.

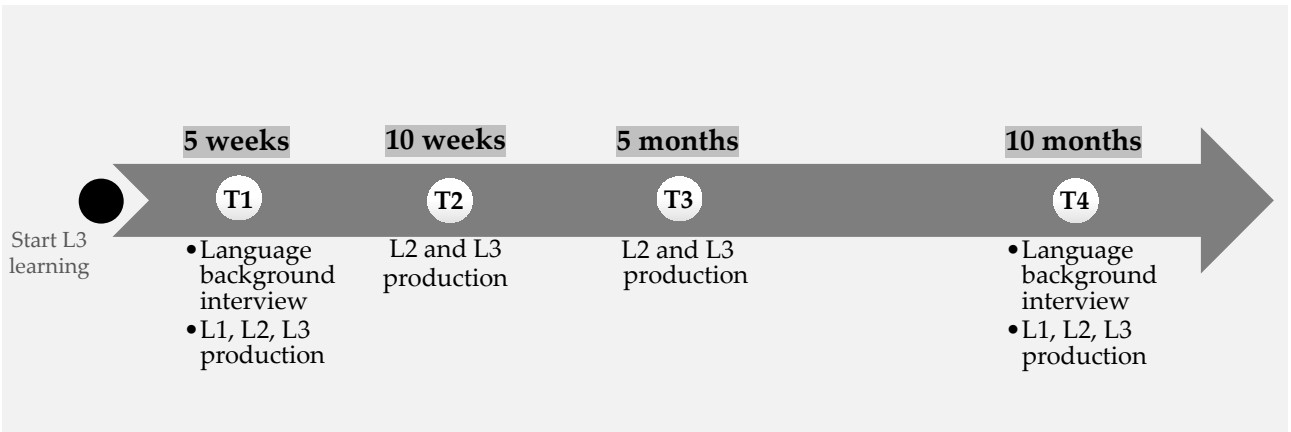

**Figure 2.** Visualization of longitudinal design.

### 2.2. Participants

Fourteen L3 learners of Polish took part in the study. They were divided into two groups according to their chronological age. Seven adolescents (aged 12–13) and seven adults (aged 21–39), all of whom spoke German as L1 and English as L2, were recorded in all of their languages. A summary of the participants' profiles is given in Table 2.

The groups were matched regarding their L3 Polish input, which amounted to three hours per week. Polish was a completely new language for all participants reported on here, i.e., all heritage speakers were excluded from the final dataset, as were all early bilinguals and speakers of additional languages not tested in this study[3]. As Table 2 shows, the groups differed in their mean starting age of L2 learning, despite the fact that they all learned the

language in a formal school setting. This is because educational policies have changed over the years concerning the school year in which English is introduced as a mandatory subject. The two groups also differed in their self-assessed L2 proficiency, which ranged from an upper-intermediate to an advanced level for the adult group, while the younger learners can all be located within a lower intermediate proficiency level. No objective measure of language proficiency was employed, as it is not a main factor of interest here.

**Table 2.** Participants' language learning profiles.

| Group | N | Age (Mean) | L1 | L2 (Mean Age of Learning; Proficiency) | L3 |
|---|---|---|---|---|---|
| Adolescents | 7 | 12–13 (12.2) | German | English (6.5; lower intermediate) | Polish |
| Adults | 7 | 21–39 (26.4) | German | English (9.4; upper intermediate to advanced) | Polish |

The adult participants were recruited in community college and university language classes, and they received a small financial compensation for participating in the study. Their reasons for learning Polish ranged from having a Polish(-descent) partner to plans of taking part in a Polish–German Erasmus exchange. The younger L3 learners were recruited at a school located in Germany within close proximity to the Polish border. They had chosen Polish as a new subject (over French) in school. They also had four 45-min English lessons (a total of three hours per week) alongside Polish lessons throughout the time span of the research project. All participants (and the adolescents' parents) provided written consent prior to the first recording session as well as verbal consent at each testing time.

*2.3. Tasks*

Three production tasks were administered to collect speech samples from the participants using the built-in microphone of a Zoom H4N Handy Recorder (Zoom Corporation, Tokyo, Japan) in a quiet room. Each of them was specifically designed to elicit several features of interest without using written words in an attempt to reduce the chance of potential orthography effects. Moreover, the items that were to be elicited were carefully chosen to approximate the learners' proficiency levels, which was naturally most challenging for their L3 Polish. Therefore, at the first testing time, the participants were only recorded conducting an imitation task in their L3. At the other testing times as well as in the other two languages, they completed all three production tasks.

In the self-paced delayed repetition task, the participants heard each target stimulus consisting of an item embedded within a carrier phrase ("I say X again") followed by a distractor phrase ("What do you say?"). After a prompt (e.g., 'And what do you say?' in English) spoken by a different speaker, the learners repeated the entire first phrase (with the inter-stimulus interval set at 1000 ms). The delayed repetition task was selected as it effectively elicits speech production even in low-proficiency speakers and reduces the risk of direct imitation by including a distractor phrase.

The second task was a self-paced picture naming, in which the learners were asked to identify a number of depicted objects and actions. These were from a range of categories that language learners are typically exposed to at the earlier stages of language learning, such as colors, animals, and foods.

Finally, the participants were presented with an 8-frame picture story they were asked to tell (a different one for each language). Again, the picture stories contained a number of objects and actions suitable to elicit the items of interest, and the research assistant administering the task made sure to guide and prompt the participant to name some of them after the participants' first rendition if necessary.

*2.4. Data Analysis*

2.4.1. Acoustic Analysis

All items starting with voiceless initial stops /p,t,k/ (followed by a vowel) were included in the acoustic analysis of voice-onset time (N = 707 for /p/, N = 456 for /t/, N = 863 for /k/). The interval between the stop release and the beginning of regular vocal fold vibration for the production of the following vowel was determined spectrographically and the boundaries were marked manually in the audio software PRAAT 6.2.03 (Boersma and Weenink 2021), in accordance with criteria well established in previous VOT studies. The main criterion for determining the release burst was the "abrupt change in overall spectrum" (Lisker and Abramson 1964, p. 389) in the waveform indicating aperiodic noise, and the onset of voicing was determined to be at the first upward zero crossing at the start of periodicity in the waveform. A PRAAT script (Stewart 2014) was employed to automatically extract the duration of the marked intervals, i.e., the voice-onset time for every individual item as well as the following vowel, in milliseconds.

2.4.2. Statistical Analysis

Generalized additive mixed models (GAMM) were used to analyze the data statistically. This was performed using the *gam()* function from the mgcv package (Wood 2017) in R (R Development Core Team 2019). Pfenninger (2021, p. 9) elucidates the advantages of performing process-oriented analyses with the help of generalized additive mixed modelling. For instance, it is well-suited to describing the iterative nature of learning processes and model complex nonlinear trajectories. Due to the different number of testing times, two separate models were created for the L1 and the L2 data. The effects investigated in the models were *Testing time* (T1 and T4 for L1; T1, T2, T3, T4 for L2), *Group* (adolescents, adults), *Task* (delayed repetition, picture naming, storytelling), *Stop* (/p/, /t/, /k/) *Speech context* (connected and non-connected speech), and *Duration of the following vowel*. The resulting mixed effect structure is presented in Table 3. The categorical variables *Task, Stop*, and *Speech context* were included as linear factors with a nested interaction with *Group*, in order to see if they had a different effect on the two learner groups. As there are only two testing times for the L1 data, *Testing time* was added in as a smooth only in the L2 model (with a separate smooth for each group), and as a linear effect in the L1 model. Tensor product smooths for *Vowel duration* and *Testing time* were included in both models, providing a natural way of representing smooth interaction terms that operate on different scales. *Participant* and *Word* were added as random smooths in both cases. For the L2 data, a scaled t distribution was selected, as this improved the model fit significantly.

**Table 3.** Mixed effects structure in L1 and L2 models.

| | | L1 Model[4] | L2 Model[5] |
|---|---|---|---|
| **Dependent Variable** | | L1 VOT duration (in ms) | L2 VOT duration (in ms) |
| **Independent Variables** | Random intercepts | s(Participant) s(Word) | s(Participant) s(Word) |
| | Categorical predictors and interactions | Group/Task Group/Speech context Group/Stop Group/Testing time | Group/Task Group/Speech context Group/Stop |
| | Smooths and tensors | te(Vowel duration, Testing time, by = Group) | te(Vowel duration, Testing time, by = Group) s(Testing time, by = Group) |

It may appear surprising that no L3 data are entered into the models as a possible predictor of L1 or L2 VOT to identify regressive CLI. This decision is based on the findings of previous studies that CLI can manifest in different ways that may not look like the most

straightforward option of similar values or merging categories. Therefore, the mixed effects models mainly serve to detect if there are any significant changes over time in both of the groups. If this is the case, they will be further interpreted based on descriptive statistics. Individual trajectories will also be analyzed descriptively, as other studies in the field with small numbers of participants have (e.g., Kopečková et al. 2016; Cabrelli 2016).

## 3. Results

The results section is structured as follows. To provide a first overview of the data, it begins with a brief descriptive presentation of the participants' VOT productions in all three languages at the four testing times on the group level. After that, L1 and L2 group results are modelled separately (due to the difference in testing times), followed by a presentation of the individual trajectories in all languages.

### 3.1. Overview of Group Results

Table 4 lists the mean VOTs measured for the three stops in all of the participants' languages at each testing time. For L1 and L2, the learners' realizations roughly correspond to the hierarchical duration pattern according to place of articulation (/p/ < /t/ < /k/). However, there is no such pattern for the L3 in either of the groups. Another interesting observation from these L3 mean values is that they all lie outside of the lower range of VOT values that would be expected for Polish.

**Table 4.** VOT means (SD) per stop in ms for the two groups at each testing time in all languages. * Note that the total means are not the simple average of the means listed here, but are 'weighted', i.e., take the number of productions of each stop into account.

| Language | | L1 | | L2 | | | | L3 | | | |
|---|---|---|---|---|---|---|---|---|---|---|---|
| Testing Time | | T1 | T4 | T1 | T2 | T3 | T4 | T1 | T2 | T3 | T4 |
| Adolescents | /p/ | 58 (18) | 49 (2) | 59 (46) | 54 (27) | 63 (31) | 52 (30) | 73 (33) | 105 (43) | 103 (47) | 97 (42) |
| | /t/ | 66 (32) | N/A (-) | 75 (32) | 64 (22) | 79 (25) | 79 (28) | N/A (-) | 83 (12) | 81 (36) | 95 (36) |
| | /k/ | 68 (22) | 85 (32) | 78 (29) | 73 (25) | 79 (32) | 83 (32) | 91 (47) | 77 (29) | 77 (36) | 79 (43) |
| | *mean** | 67 | 84 | 69 | 62 | 75 | 73 | 83 | 91 | 87 | 89 |
| Adults | /p/ | 43 (13) | 48 (18) | 56 (22) | 54 (21) | 60 (25) | 60 (21) | 66 (19) | 69 (28) | 70 (28) | 69 (34) |
| | /t/ | N/A (-) | 51 (16) | 79 (23) | 81 (25) | 74 (26) | 75 (29) | 49 (14) | 80 (32) | 64 (19) | 64 (17) |
| | /k/ | 69 (20) | 68 (19) | 76 (19) | 74 (21) | 73 (20) | 72 (24) | 63 (24) | 67 (21) | 67 (27) | 62 (18) |
| | *mean** | 67 | 65 | 70 | 69 | 69 | 68 | 64 | 71 | 67 | 66 |

Due to the complexity of the dataset with the number of testing times and languages involved, from now on the three stops are lumped together in visualizations, such as Figure 3, which shows the group VOT means per language at each testing times with all three stops combined. The graph reveals some major group differences: While the VOT means measured in the three languages are virtually aligned at every testing time for the adults, the adolescents' values for the three languages are much more spread out (except for an overlap of their L1 and L2 at T1). Moreover, as the error bars demonstrate, there seems to be more inter-individual variation in the adolescent group, which is most pronounced in the L3. What is more, there is little observable change for the adults' languages on the group level (other than a slight increase and decrease in the L3). The adolescents, on the other hand, exhibit more pronounced changes over time in all of their languages, including the L1, for which mean VOTs increased by 17 ms from T1 to T4. The models in the following sections will determine whether these group trends are statistically significant.

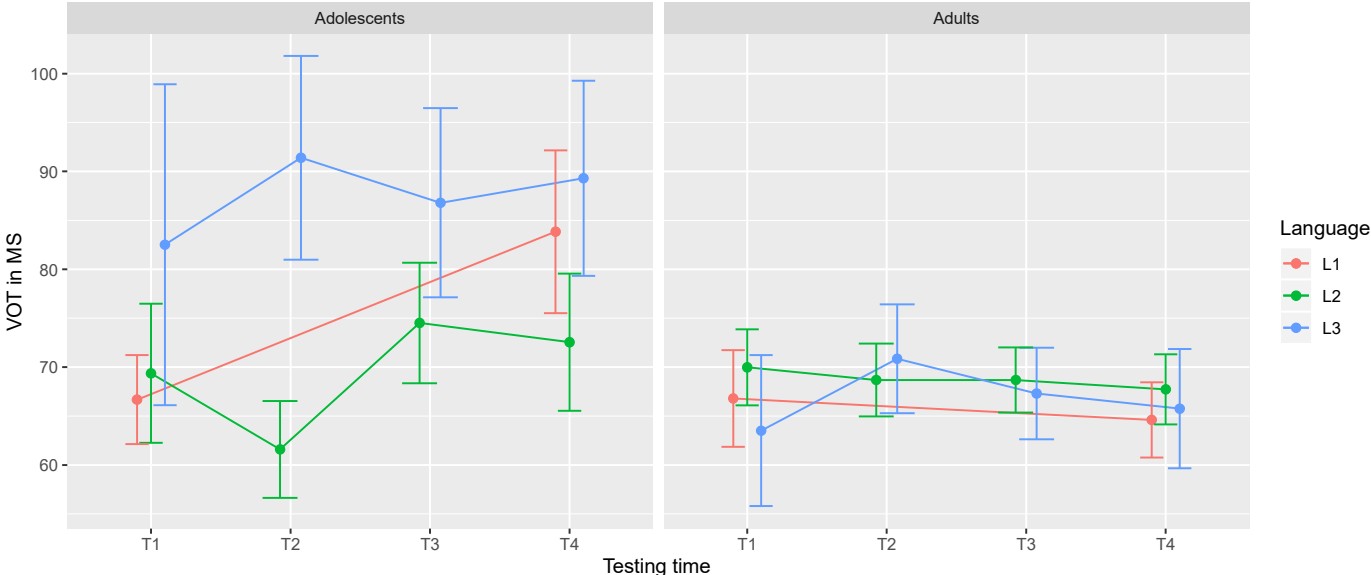

**Figure 3.** Group VOT means per language at each testing time (all stops combined).

### 3.2. L1 Development

After plotting and inspecting the dataset descriptively, the L1 VOTs were fitted to a generalized additive mixed model to assess the effect of testing time, stop, task type, speech context, and vowel duration on both groups' VOT productions. Table 5 displays the model output. Among the categorical factors, there are several significant interactions between *Group* and *Stop*, indicating that both groups produced significantly lower VOTs for /p/ than for /k/. The nested interaction for the adolescents with *Testing time* denotes a significant increase in VOTs for this group from T1 to T4. There was no significant development over time for the adults. Furthermore, neither of the groups showed a main effect of task, speech context, or duration of the following vowel. The random smooths for *Participant* and *Word* were significant.

**Table 5.** GAMM output for the L1 VOTs. Reference values are determined alphabetically (testing time 1, stop /k/, task DR, speech context connected).

| Parametric Coefficients | Estimate | Std. Error | *t* Value | *p* |
|---|---|---|---|---|
| (Intercept) | 66.6745 | 7.0270 | 9.488 | <0.001 *** |
| GroupAdults | 0.4375 | 7.1574 | 0.061 | 0.951 |
| GroupAdolescents:TaskPN | 1.7444 | 9.2185 | 0.189 | 0.850 |
| GroupAdults:TaskPN | −3.9345 | 9.3670 | −0.420 | 0.675 |
| GroupAdolescents:TaskST | −6.4040 | 7.4912 | −0.855 | 0.393 |
| GroupAdults:TaskST | −1.6717 | 7.6980 | −0.217 | 0.828 |
| GroupAdolescents:ContextNON | 8.1827 | 6.1550 | 1.329 | 0.185 |
| GroupAdults:ContextNON | 1.8098 | 6.2760 | 0.288 | 0.773 |
| GroupAdolescents:Stopp | −16.0890 | 8.0505 | −1.999 | 0.047 * |
| GroupAdults:Stopp | −20.9495 | 6.9364 | −3.020 | 0.003 ** |
| GroupAdolescents:Stopt | 8.1732 | 11.2323 | 0.728 | 0.467 |
| GroupAdults:Stopt | −10.2198 | 10.6449 | −0.960 | 0.338 |
| GroupAdolescents:Testing_time4 | 11.7948 | 3.3689 | 3.501 | <0.001 *** |
| GroupAdults:Testing_time4 | 0.4778 | 3.2971 | 0.145 | 0.885 |

| Approximate significance of smooth term | edf | Ref. df | F | *p* |
|---|---|---|---|---|
| s(Participant) | 9.8822 | 12 | 4.766 | <0.001 *** |
| s(Word) | 15.6983 | 40 | 1.756 | <0.001 *** |
| te(Vowel_duration,Testing_time ):GroupAdolescents | 0.1883 | 8 | 0.029 | 0.361 |
| te(Vowel_duration,Testing_time):GroupAdults | 0.6535 | 8 | 0.150 | 0.218 |

**Table 5.** *Cont.*

| | | | |
|---|---|---|---|
| R-sq. (adj) | 0.406 | Deviance explained | 47.7% |
| -REML | 1412.7 | Scale estimate | 344.71 |
| n | 329 | | |

*Note:* edf = effective degrees of freedom; Ref. df = reference degrees of freedom; R-sq. (adj) = R-squared adjusted; -REML = restricted maximum likelihood; significance levels: '*' < 0.05, '**' < 0.01, '***' < 0.001.

Figure 4 shows the mean L1 VOTs of the individual participants at T1 and T4. Mirroring the trends identified in the group data, there was only one adolescent whose values did not increase over time. The adults' data, on the other hand, are not quite as uniform. Even though changes are not drastic for any of the adults, two produced longer VOTs, two produced shorter VOTs, and three had nearly identical means at the two testing times.

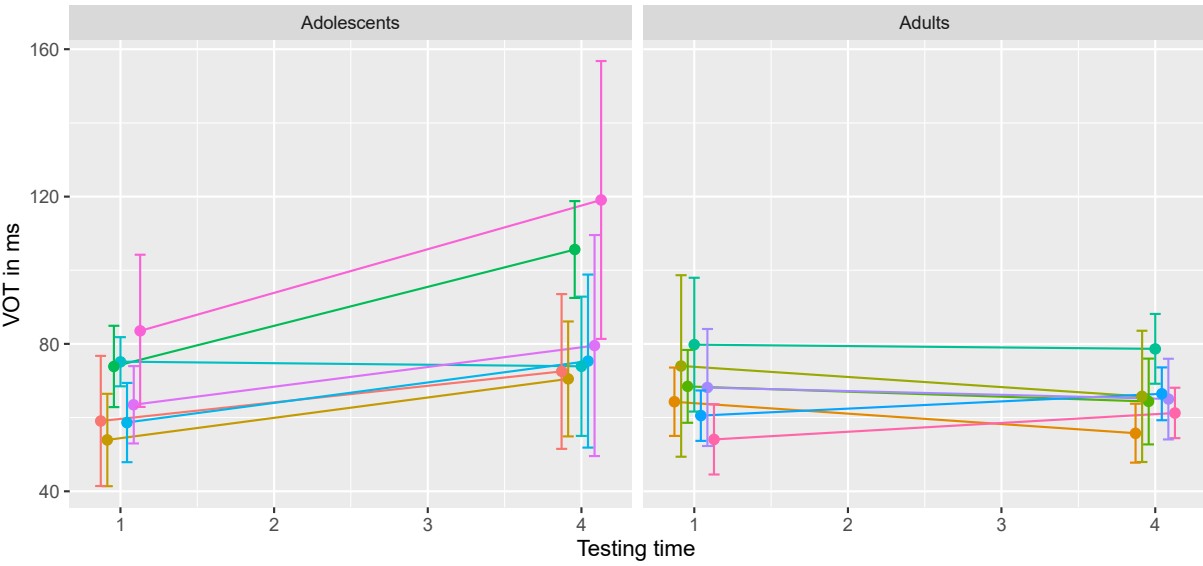

**Figure 4.** Mean VOT of individual participants in L1 (all stops combined).

### 3.3. L2 Development

Table 6 shows the L2 model output. Place of articulation emerged as significant for both groups (indicating significantly shorter VOTs for /p/ than for /k/). There were no significant effects of *Task* or *Speech context.* However, the smooth for *Vowel duration* (in interaction with *Testing time*) was significant. With regard to development over time, the smooth for *Testing time* was significant for the adolescent group, but not for the adults.

**Table 6.** GAMM output for the L2 VOTs. Reference values are determined alphabetically (stop /k/, task DR, speech context connected).

| Parametric Coefficients | Estimate | Std. Error | *t* Value | *p* |
|---|---|---|---|---|
| (Intercept) | 74.192 | 6.225 | 11.919 | <0.001 *** |
| GroupAdults | 5.317 | 7.444 | 0.714 | 0.475 |
| GroupAdolescents:TaskPN | −1.760 | 5.816 | −0.303 | 0.762 |
| GroupAdults:TaskPN | −5.126 | 4.980 | −1.029 | 0.303 |
| GroupAdolescents:TaskST | 2.906 | 3.905 | 0.744 | 0.457 |
| GroupAdults:TaskST | −5.925 | 3.326 | −1.781 | 0.075 |
| GroupAdolescents:ContextNON | 4.128 | 2.756 | 1.498 | 0.134 |
| GroupAdults:ContextNON | 1.028 | 2.639 | 0.389 | 0.697 |
| GroupAdolescents:Stopp | −16.380 | 3.658 | −4.478 | <0.001 *** |
| GroupAdults:Stopp | −13.368 | 2.939 | −4.549 | <0.001 *** |

**Table 6.** *Cont.*

| Approximate significance of smooth term | | edf | Ref. df | F | p |
|---|---|---|---|---|---|
| GroupAdolescents:Stopt | | −2.101 | 4.011 | −0.524 | 0.600 |
| GroupAdults:Stopt | | 4.917 | 3.405 | 1.444 | 0.149 |
| **Approximate significance of smooth term** | | **edf** | **Ref. df** | **F** | **p** |
| s(Participant) | | 11.539 | 12.000 | 340.67 | <0.001 *** |
| s(Word) | | 41.451 | 115.000 | 227.04 | <0.001 *** |
| te(Vowel_duration,Testing_time):GroupAdolescents | | 3.396 | 12.000 | 135.11 | <0.001 *** |
| te(Vowel_duration,Testing_time):GroupAdults | | 3.979 | 12.000 | 329.58 | <0.001 *** |
| s(Testing_time):GroupAdolescents | | 1.001 | 1.001 | 10.42 | 0.001 ** |
| s(Testing_time):GroupAdults | | 1.365 | 1.620 | 0.28 | 0.742 |
| R-sq. (adj) | 0.395 | Deviance explained | | 40.3% | |
| -REML | 5152.7 | Scale estimate | | 1 | |
| n | 1153 | | | | |

*Note:* edf = effective degrees of freedom; Ref. df = reference degrees of freedom; R-sq. (adj) = R-squared adjusted; -REML = restricted maximum likelihood; significance levels: '*' < 0.05, '**' < 0.01, '***' < 0.001.

Moving from group modelling to a closer inspection of individual trajectories, Figure 5 demonstrates the L2 VOT mean scores for every participant from T1 to T4. It is evident at first sight that the adults' means appear to be much more uniform than the adolescents' means. Although the VOTs of some of the adults *do* change in non-negligible manners over time, most means in this group roughly remain within the range of 60 to 85 ms that is expected for English. The changes in the adolescents' group are more pronounced, albeit in the absence of a detectable group-specific pattern. In order to be able to qualitatively interpret the development of both groups in a more meaningful way, the following section considers individual trajectories through the lens of the speakers' entire repertoire.

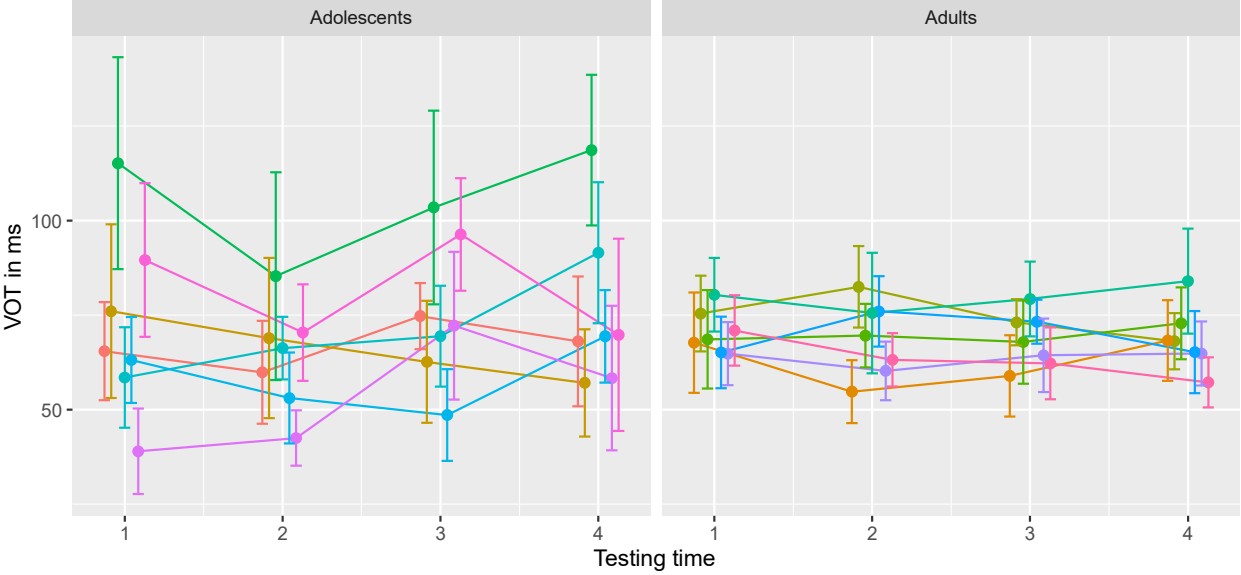

**Figure 5.** Mean VOT of individual participants in L2 (all stops combined).

### 3.4. Individual Trajectories—Entire System

Figure 6 shows every participant's VOT means in the three languages at all testing times. This visualization uncovers the sheer variety of combinations of how the languages can relate to one another over time. Even though identifying patterns that apply to multiple participants across the board seems challenging, this short list is an attempt to do so:

- **Similar values for L1 and L3 at T1** (e.g., DOSC23, MESC03, SISC11 and SMSC15 in the adolescents' group; no one in the adults' group): Initially, VOTs in the new language are realized in an L1-like fashion.
- **Similar (but non-overlapping) trajectories for L2 and L3** (e.g., JUEB20, MASC05 and SISC11 in the adolescents' group; REBA03 and SYLU08 in the adults' group): Rises or falls within one language are reflected in the other language as well.
- **Similar values for L1 and L2 at both T1 and T4** (e.g., BISC14 and MESC03 in the adolescents' group; EDMU06, LYBO24, REBA03, and ROGI18 in the adults' group): English and German, the two languages that would be expected to behave similarly, are in fact produced with overlapping VOT values at T1 and T4.

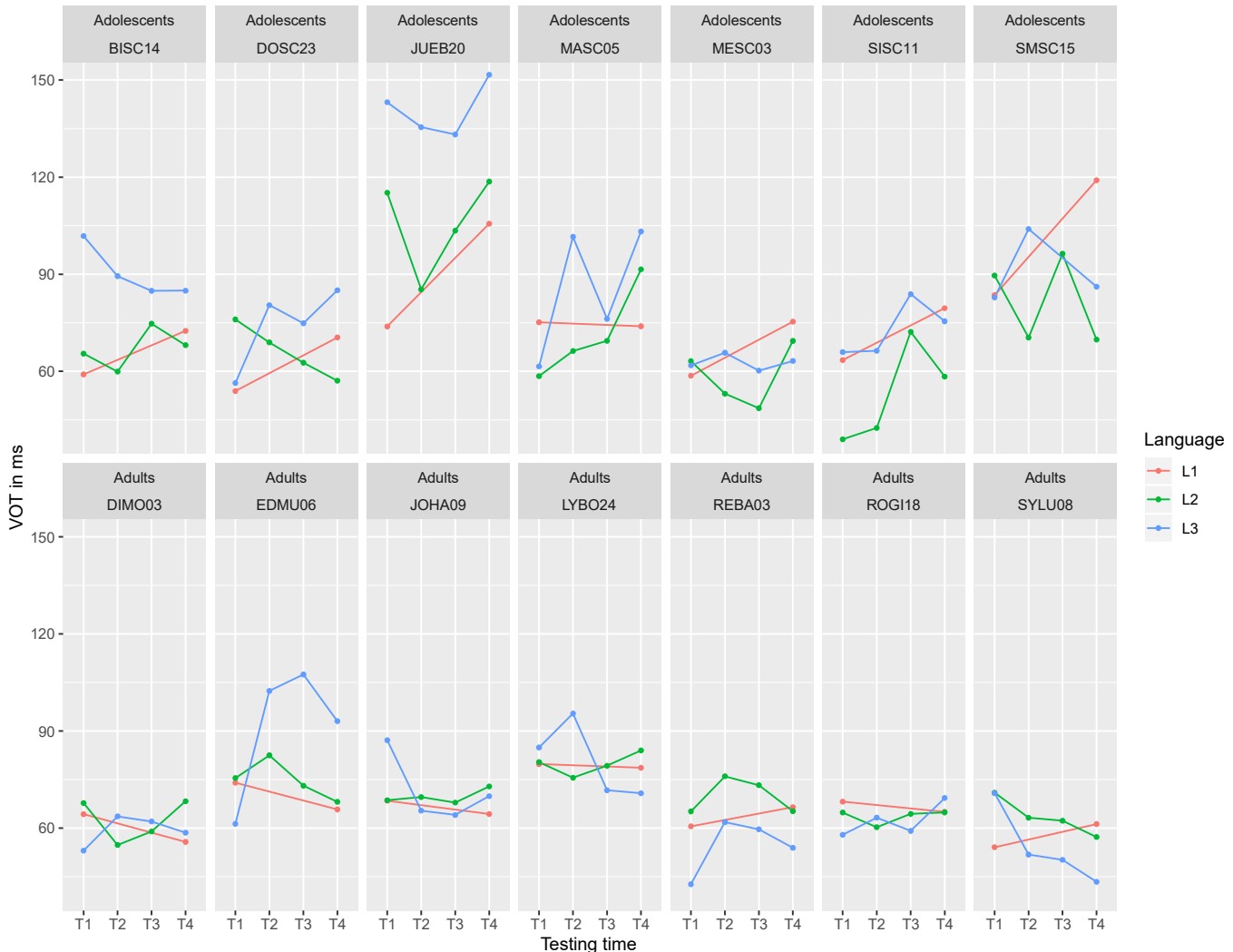

**Figure 6.** Mean VOT of individual participants in all languages (all stops combined). The dots are connected with lines for better visibility; however, especially in the case of the L1 data with only two testing times, one has to be careful not to assume that this means there is a linear development.

Homing in on the tentative interpretation of the learners' L3 as source of regressive CLI, Table 7 roughly bins every participant's L1 and L2 trajectory compared against their L3 development according to four categorizations: (1) *Little to no change*, (2) *Change towards the L3*, (3) *Change away from the L3*, (4) *Development in alignment with the L3* (for the L2 only). Regarding the learners' L1, the table shows that the VOT means of four out of seven adolescents changed towards the L3, while this was only the case for one adult. Two participants in each group produced L1 VOTs deflecting away from their L3 values over

time. The majority of adults (four out of seven) and only one adolescent did not experience a change comparing T1 and T4.

**Table 7.** Approximate categorization of each participants' L1 and L2 trajectory in light of their L3 values.

| Language | Group | Little to no Change | Change Towards L3 | Change Away from L3 | L2 Only: Development in Alignment with L3 |
|---|---|---|---|---|---|
| **L1** | *Adolescents* | MASC05 | BISC14 DOSC23 JUEB20 SISC11 | MESC03 SMSC15 | N/A |
| | *Adults* | JOHA09 LYBO24 REBA03 ROGI18 | DIMO03 | EDMU06 SYLU08 | N/A |
| **L2** | *Adolescents* | - | BISC14 (first deflection, then approximation) MESC03 (first deflection, then approximation) | DOSC23 (first approximation, then deflection) SMSC15 (deflection, approximation, deflection) | JUEB20 MASC05 SISC11 |
| | *Adults* | JOHA09 ROGI18 | - | DIMO03 (deflection, approximation, deflection) EDMU06 LYBO24 (deflection, approximation, deflection) | REBA03 SYLU08 |

For the L2, the previously mentioned fourth scenario of aligned trajectories emerges (thanks to the four testing times in both of these languages), where L2 and L3 means did not overlap, but appear to have developed in parallel. This could be observed for three adolescents. The remaining adolescents either showed a change towards or away from the L3 (two learners each). In the adult group, the VOTs of two learners did not change much over the four testing times. For those whose did, two had aligned L2–L3 trajectories and three produced L2 VOTs that seemed to change away from their L3 VOTs over time. In both groups, such deflections were mostly preceded by an initial approximation of L2 and L3 values.

In summary, this qualitative look at the individual data confirms some of the tendencies found by the group models; for example, more changes in both L1 and L2 for the adolescents than for the adults. However, it also reveals some intriguing more fine-grained observations with respect to the *kind* of changes taking place in both groups. For instance, the scenario of a background language approximating or developing towards L3 values seems more common in the adolescent group (four for L1, two for L2; compared to one adult for L1 and none for L2). Connected L2–L3 trajectories, on the other hand, appear to be present to a similar extent in both groups. Furthermore, from this table it seems like there are more changes to the adults' L2 than their L1, but since no L1 data were collected at T2 and T3, this assumption cannot be fully corroborated. L2 trajectories like the ones of DIMO03 or REBA03 demonstrate just how informative it is to include multiple testing times—had they only been tested at T1 and T4, it would have looked like there were no changes over time. However, taking into account T2 and T3 paints a different picture, and can ultimately tell us much more about how languages develop alongside each other over time. This and other aspects are now addressed in the discussion.

## 4. Discussion

The overarching research question of this study was whether a learner's background languages, in this case their L1 and L2, would change with increasing exposure to a new language. Another layer to this main research aim was added by testing two different age groups of L3 learners with the same language combination and comparable language learning histories. Both chronological age as well as other age-*related* aspects were expected to play a role in the potential occurrence of language changes: the former in the case of L1, as studies have shown that an adolescent's L1 is still developing and thus more malleable (albeit not to the same extent as their L2), and the latter more so in the case of the learners' L2. The fact that the adult learners in this study have been exposed to the L2 for much longer (and are thus more proficient in it) and do not receive formal instruction in it anymore is a by-product of them being older, i.e., an age-related aspect. Concerning language status, it was hypothesized that for both learner groups, the L2 would be more malleable and thus more susceptible to L3 influence than their L1, based on previous research on regressive phonological CLI as well as the Phonological Permeability Hypothesis (Cabrelli 2016). Regarding group differences, it was predicted that relatively speaking, the adults' languages would exhibit fewer changes than the adolescents' languages. Further, significant individual variation and non-linear trajectories were expected for all languages and speakers. Since these research questions are somewhat entangled, it is impossible to answer one without the others, so they will now be discussed jointly.

Acknowledging evidence from other studies showing that CLI can take on many forms (i.e., not only resulting in approximating values, but also in a deflection away from the other language), the first step in answering the research questions was to determine statistically if significant changes of *any* kind took place in the groups' L1 and L2 over the first 10 months of L3 learning. This was then followed by an investigation of the individual learners' trajectories.

On the group level, substantial changes over time were only found in the adolescents' group (for both languages), confirming the second hypothesis that their L1 would be more malleable than that of the adults' due to maturational effects, and that their L2 would be more malleable due to them still actively learning the language, which also renders it more susceptible to influence. In their L1 German, VOTs increased significantly comparing T1 and T4, resulting in untypically long VOTs for some participants. However, this is not to be interpreted as a linear increase over time. In fact, it is much more likely that any measurements in between these two testing times would not have fallen quite on the imaginary trajectory line in between T1 and T4. This is indeed one of the great limitations of the study. Due to the expectation prior to designing the study that there would not be much change to observe in the learners' L1, combined with a testing battery that was already long as it was, it was decided only to test the L1 at the beginning and the end of the testing period. The first research question regarding which of the background languages would be affected more can, therefore, not be answered conclusively for either of the groups. To allow greater comparability between the languages, future longitudinal research into regressive CLI should include more testing times even for the speakers' L1, as there indeed appears to be a chance of change over time (at least for younger learners). Still, the present results indicate a greater degree of reorganization of the adolescents' L1 and L2 than that of the adults' when it comes to their production of VOT over time, which is broadly in line with CDST theorizing: the interactions between the learners' characteristics, such as the initial conditions (the state of development in the L1 and L2) and the availability of resources (such as attention, language input, and the opportunity to use the language) are of a different nature for the two learner groups, leading to different developmental patterns for them.

A descriptive look into the learners' individual trajectories further uncovers much variation within both groups. As shown in Figure 6, for most learners, the relationship between their languages changed over time. As a result, each person's multilingual trajectory is so unique that categorizing them according to common patterns proved

difficult. Still, a few broad similarities were identified. For instance, a number of learners produced similar VOT means for their L1 and L2, the two aspirating languages, both at the beginning and the end of the year. This especially applies to the adults. In fact, only one adult clearly keeps their L1 and L2 apart at T1 (SYLU08 with a 17 ms difference between the two). By contrast, five out of the seven adolescents produced L1 and L2 VOT means that differ by 15 ms or more at one or both testing times. Another observation that is especially noteworthy is that of parallel L2–L3 trajectories found for some speakers in both groups. Whereas this is not the only kind of relationship for connected trajectories that is conceivable according to CDST theorizing, it neatly demonstrates the interconnectedness of these two subsystems. Additionally, it illustrates an important point frequently made at the intersection of CDST and attrition research, but sometimes neglected in CLI studies: The existence of regressive CLI (and perhaps even any kind of CLI) implies two coexisting, interdependent processes, namely the acquisition of new structures on the one hand, and attrition or modification of already existing language structures on the other hand (see, e.g., De Bot and Larsen-Freeman 2011, p. 6; Cabrelli 2016, p. 3). Understanding regressive CLI as a one-way road thus likely results in unrealistic expectations and inaccurate descriptions. In other words, in many cases it will be difficult to interpret CLI effects and their directionality with certainty. The implied expectation of clearly identifiable senders and receivers of CLI that many studies start out with (including this one) is challenged by the learners' trajectories encountered here, and not only in the cases of the parallel L2–L3 developmental paths. Further, for those learners whose background languages deflect away from or move towards L3 values (or towards each other), this can best be interpreted as mutual or bi-directional influence.

Moreover, it should also be mentioned that the finding of overall little development over time for some learners in the adult group does not necessarily contradict CDST assumptions of a complex system. As mentioned previously, systems can also enter so-called attractor states, in which it "takes considerable amount of energy to make a system break away [ . . . ] and move on" (De Bot and Larsen-Freeman 2011, p. 15). It is possible that adult learners need a different amount of L3 exposure or relevant phonetic input to reorganize their phonological system. Another issue is that of selecting appropriate time frames: It may be that for the adults, one needs a different time window of observations than that for the adolescents to capture the developmental changes in their background languages. In an earlier publication with the same group of learners on the L2/L3 perceptual development of the /v-w/ contrast (Nelson 2020), the most inter-speaker variation in the adults' L2 was found at T2, i.e., 10 weeks into L3 learning. While individual data rather than group means would be necessary to fully make this connection, this might be an indication that the reorganization of the adult system happened earlier (albeit less pervasive than for the younger learners). The collection of dense data, which is a process-oriented research design typically employed in the CDST framework, may provide insight into this hypothesis. Learner groups or individuals could participate in numerous densely spaced testing times, i.e., every week for the first three months of L3 learning, followed by analyses that focus on the description of the dynamic development of all their languages. Such a design would also be more suitable to truly shed light on the hypothesis that major changes in the system are preceded by a high degree of variability (Evans and Larsen-Freeman 2020; van Dijk and van Geert 2007). In the field of phonology, dense data collections are very rare—understandably so, since both collection and analysis are very time-consuming, especially if they are conducted in all of a speaker's languages. However, the time investment could be worth it, as such research could offer invaluable insights into multilingual language development.

With regard to the PPH whose predictions were partly tested here (albeit with a somewhat different methodological setup than envisioned by its proponents), the findings both confirm and contradict aspects of it at the same time. The hypothesis that there would be more change in the adolescents' languages, deduced from the PPH's reasoning about the effect of language stability, could be confirmed. However, the PPH also predicts that

the learners' L2 would be more vulnerable to L3 influence than the L1, which was not the case. Cabrelli's (2016) initial study that delivered such evidence, at least on the level of production, featured learners with two non-native languages that were very similar (Spanish and Brazilian Portuguese). In the absence of such linguistic proximity between the L3 and either of the background languages (i.e., between Polish and German, and Polish and English), it is unclear under what conditions and time frames regressive CLI can be expected for adult learners in an instructed learning context. It is also possible that factors other than language status play a significant role, as for instance the language of instruction or classmates in the L3 classroom. In research on forward CLI (Llama and López-Morelos 2016) it has been reported that heritage speakers produced non-target VOT structures where they could have transferred correct ones from their heritage language, probably due to the fact that their non-heritage classmates with their L1-accented speech were their main source of L3 input. It is conceivable that such variables can also be important for regressive CLI. For example, interplay between two of a speaker's languages could be encouraged if they are frequently used alongside each other in the classroom.

   In addition, it has to be mentioned that some sense of uncertainty generally remains with the interpretation of any kind of changes within a language as a result of CLI in this study. It is difficult to rule out the possibility that other factors could also have contributed to the observed developments, especially given the drastic variability found among the adolescents. Even the timing of the testing sessions as well as the recording situation itself may have had a different impact on the two age groups. Although much care was taken by the research assistants to create a comfortable and enjoyable atmosphere for all participants, it might have been perceived as an examination-like setting by the less-proficient adolescent learners more so than the adults, and the resulting nervousness may have had an influence on their pronunciation. This is a general issue for studies comparing speakers of vastly different ages, even when other external factors are carefully controlled for. A possible, albeit imperfect solution that would enable us to assess with greater confidence whether observed changes are in fact CLI-induced, might be the inclusion of control groups. For example, speakers without the L3 input the experimental group receives but with otherwise comparable profiles could be recorded over the course of a year. While this sounds like an optimal solution in theory, there are a number of challenges one faces when recruiting such a control group, especially when it comes to the younger learners. Unless they attend a different school form (which would likely make the groups incomparable on other levels associated with the choice of school form in a tiered system, such as it exists in Germany, as for example their socio-economic background), most 11–13-year-olds in Germany are required to start learning a third language in Year 6 (typically French, Spanish, or Dutch). This would thus leave us with comparing groups with differing L3s (as opposed to one with and one without L3 experience), as has been conducted in previous studies on forward transfer (e.g., Wrembel 2014; Sypiańska 2016; Dittmers et al. 2017). Whereas this might be interesting with regard to other research questions, it might not deliver more conclusive evidence either when faced with the challenge of identifying regressive CLI, considering that such changes may exhibit in a variety of ways. A more suitable option would be starting to trace the same L3 learners' development for a while *before* the onset of L3 exposure to then be able to fully assess the impact learning a new language has on the entire system. Such a method would also be more in line with CDST theorizing, as it would be more informative for the exploration of individual learning trajectories.

   A further limitation of this study may also lie in the conceptualization of age as a categorial variable inherent to the group comparison employed here. This has been criticized previously for studies on the effect of age of onset and L2 'ultimate attainment' with typical groups of early and late starters, as such a methodology does not, in fact, allow us to see if there is an abrupt decline in learning abilities at a certain age or rather a more gradual one (see, e.g., Muñoz and Singleton 2011; Hakuta et al. 2003). Moreover, the fact that there is much disagreement among researchers as to where such a cut-off point would be casts doubt on the general idea of a critical period and therefore on the

conceptualization of age as a categorical variable in language learning studies. While this study provides some evidence for developmental differences between the adolescent and adult participants, a more fine-grained analysis of age(-related) effects would be desirable. Future CLI studies interested in age or age-related effects could thus recruit a greater variety of multilinguals from early adolescence (or even childhood) through adulthood. Instead of employing a group comparison, age could then be treated as continuous variable.

To summarize, the present study has shown that the multilingual language development over the first year of L3 learning differs for adolescents and adults. While the adolescents exhibited significant changes in both of their background languages, the adults did not, at least not on the group level. However, as expected, the investigation of individual trajectories uncovered much inter- and intra-speaker variation and a variety of ways in which the development of a speaker's languages can be connected. Future research on regressive CLI is encouraged to continue with longitudinal investigations, including more densely spaced testing times if possible, as well as other potentially influential factors beyond language status and age.

**Funding:** This research was supported by the German Research Foundation (Deutsche Forschungsgesellschaft) in connection with the project entitled "Cross-linguistic influence in the acquisition of phonology and phonetics by multilingual children and adults" (April 2017–February 2020), grant number GU 548/11-1 | KO 5158/4-1.

**Institutional Review Board Statement:** The present study was conducted in accordance with local legislation and the institutional requirements, and it follows both the Code of Ethics "Rules of Good Scientific Practice" of the University of Münster (2002) and The European Code of Conduct for Research Integrity (European Federation of Academies of Sciences and Humanities, 2017). In addition, for the adolescent participants, approval of the responsible education ministry and school board was obtained.

**Informed Consent Statement:** Informed consent was obtained from all subjects involved in the study.

**Data Availability Statement:** The data presented in this study are openly available in the Open Science Framework (OSF) at https://osf.io/w6n8m/ (DOI: 10.17605/OSF.IO/W6N8M) (accessed on 23 March 2022).

**Acknowledgments:** The research reported here was supported by the German Research Foundation. This grant as well as the invaluable support of Ulrike Gut and Romana Kopečková as Principal Investigators are hereby gratefully acknowledged. I am greatly indebted to the adolescent and adult learners for their enthusiastic participation. I am also grateful to the other research assistants who helped with the data collection, as well as to the reviewers and editors for their helpful comments and suggestions. Any remaining errors remain my own.

**Conflicts of Interest:** The author declares no conflict of interest.

## Notes

[1] The data for this study and the one reported in Nelson (2020) were collected within the same project, so these are the same L3 learners.

[2] It should be noted that according to CDST theorizing, language systems are unstable by definition. The PPH terminology therefore seems somewhat at odds with that conceptualization. However, given that the PPH does not presume any subsystem to ever be fully stable (which would be against CDST theorizing) but rather focuses on *relative* differences between two subsystems, the two are not incompatible.

[3] Five of the seven adults had also taken some French and Spanish lessons in secondary school and were thus tested in these languages as well. However, the findings of these additional languages are not reported here, as the focus lies on the three languages shared by all participants.

[4] The following call was used for the L1 model: L1gam <- gam(Duration ~ Group/Task + Group/Context + Group/Stop + Group/Testing_time + s(Participant, bs = "re", k = 14) + s(Word, bs = "re") + te(Vowel_duration, Testing_time, bs = "fs", m = 1, by = Group), data = L1, method = "REML").

[5] The following call was used for the L2 model: L2gam <- gam(Duration ~ Group/Task + Group/Context + Group/Stop + s(Participant, bs = "re", k = 14) + s(Word, bs = "re") + te(Vowel_duration, Testing_time, k = 4, by = Group) + s(Testing_time, k = 4, by = Group), data = L2, method = "REML", family = scat(link = "identity")).

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
