# Peer review of "Do a Learner’s Background Languages Change with Increasing Exposure to L3? Comparing the Multilingual Phonological Development of Adolescents and Adults"

_languages, doi:10.3390/languages7020078_

Round 1

Reviewer 1 Report

This is a clear report of an interesting study that contributes to the body of existing literature, and I enjoyed reading it. I especally appreciate the brave decision to focus on the complicating inclusion of L3 acquisition and its effects on L1 and L2. The choice for VOT works very well for these three languages so forms a very relevant basis of comparison. Although I would certainly be in favour of publishing this paper, I think some issues will have to be reconsidered and some polishing is required for the argumentation.

My biggest concern is the disbalance between the argumentation in the introduction and the discussion versus the method used. The introduction emphasizes the importance of a CDST perspective, in which we look at the develoment of (individual) trajectories over time. This is repeated in the discussion. The method, however, is not compatible with this position. First, the use of process studies focusing on development over time may be problematic due to the ergodicity condition (see Lowie & Verspoor, 2019 - in Language Learning). Second, the use of LMERS is absolutely find for measurements at one moment in time, but is not suitable for development over time, because it assumes linearity of relationships. Your table 4 clearly confirms the nonlinearity of the data and the variability over time. The use of General Additive Mixed Models (GAMMs) would be more suitable (see the recent work of Pfenninger (for instance in International Review of Applied Linguistics in Language Teaching). From a CDST perspective, Figure 6, showing the divergent development of the individual learners is very revealing and illustrates the ergodicity issue. 

Another issue is the way in which the paper deals with the age factor (or age-related factors?). While the authors acknowledge the CPH as a starting point for research, they nevertheless regard age as a categorical variable, while age is essentially a continuous variable. The selection of groups, still seems to use the CPH as a starting point, which has now repeatedly been shown to be problematic (see eg the recent work of David Singleton). I understand this is not easy to repair in retrospect, but a remark about this limitation would be suitable.

A third point of critique is the way in which the paper treats "stability". The assumption is that "L2 will be more vulnerable to L3 influence than the L1 due to the difference in stability" (p3). I agree, but I miss the theoretical foundation of this assumption. A central issue of CDST is that the language  system (of any language) is unstable by definition. Even though any of the embedded subsystems may show (temporary) stability manifested by a lower degree in variability, the system as a whole will never be stable. On page 8 you summarize the findings as "mixed and exhibit much individual variation". Here again the link to a CDST framework would make sense. So, assumptions about the difference in stability between L1, L2, and L3 are certainly worth exploring, but this will need a more thorough foundation in the discussion of the background literature.  This is an important point, as it is also the foundation of one of the RQs. Apart from the perspective on stability and variablility, your review of the literature on regressive CLI is a very clear and comprehensive.

Some comments on details:

  • Your Table 1 is very useful. Would there be a way to be more explicit about the findings. Some of the findings found were really clear, while others were more marginal. Can you include effect sizes or say something in general about the magnitude of the effects found?
  • You opted for four testing times with increasing time intervals. What was the rationale for your "compromise". As has been argued by Lowie (2017, "Lost in state space"), the choice of the density and frequency of longitudinal measurements is essential and requires a rationale.
  • RQ3 - I understand you cannot formulate a hypothesis, but based on CDST literature on the importance of variability it would be possible to express some expectation on this point.
  • Since there were clear difference for the different places of articulation, summarizing the VOTs of different positions leads to loss of information. Since the stongest effect appears for /p/, wouldn't it make sense to focus on /p/?
  • Since stability and variability are an important issue in the paper, it would be good to try and include error bars around the mean VOT values in Figure 5.
  • In the discussion you refer to Cabrelli (2016) and the fact that the languages in this study were typologically similar. Wouldn't you regard the languages in your own study as typologically similar? At least German and English are not very distant
  • Please thoroughly check consistency of referencing layout / documentation (APA?) and style and spelling (eg DCST --> CDST)

Reviewer 2 Report

As an overall impression, this study is original and very well carried out and reported. Relating to Cabrelli’s Phonology Permeability Hypothesis (PPH) and recent research along these lines and carrying empirical studies in this area further, this contribution is timely and innovative. It manages to include and address a range of aspects which are motivated by the PPH and are understudied (especially in combination) also in a more general context of second and third language acquisition. Such aspects are:

- a longitudinal design, especially one that starts at the very initial stage of the language being learnt (a criterion which may often be hard to meet in practice);

- a joint examination of the learners’ whole repertoire, i.e. both the language being learnt and the background languages, and really testing the background languages;

- a focus on regressive CLI;

- a comparative investigation of learners of different age groups;

- going beyond the group level and mean values to a detailed study of the results and variation at the individual level.

The process-orientation, the focus on processes of ongoing development, can also be mentioned here.

The presentation of the problem and the exposition and discussion of the results are thorough and clear throughout.

This being said, I can see a problem of uncertainty with the interpretation of the findings, which needs to be attended to. This is why I marked a need for improvements on two of the questions to be answered above. The study aims to investigate the occurrence of regressive CLI, in line with the tenets of the PPH. But, strictly speaking, how can we be sure that the results presented are really instances of CLI and not caused by other factors? This point is essential, because the interpretation of the results and hence the conclusions depend on this. What gives raise to some doubts are the very disparate results, especially for the adolescents (Figures 3 and 6  and Table 7). There is a drastic variation both between individuals and between times of testing. The disproportionately high VOT values for L3 Polish for most of the adolescents is also remarkable. Effects of this kind when a new language enters the repertoire may fit with the Complex Dynamic Systems Theory, to which the author(s) also refer(s). But adopting this view is not the same as a proof that (only) CLI is at play here. Suppose, for example, that factors in the test situation have played a part. Especially with the adolescents who were less proficient in their L2 English and were still doing tests in a school atmosphere, there might have been factors in the test situation, like nervousness, shaky pronunciation skills, over-articulation, or random incidents. If so, this might also have played a part for the differences between adolescents and adults. A speculative scenario, certainly, but conceivable. There is no independent evidence for a CLI interpretation. (Or is there? Can, e.g., comparable results elsewhere provide some support?) In any case, other potential factors than CLI ought to be controlled for as far as possible, and the problem should be acknowledged, and necessary reservations made.

Some minor points:

page 2, line 95: micro- and macro-level. What does this mean?

p.8, l. 358: “even in an ostensibly native-like L2 is more vulnerable to L3 influence than an L1”. This quotation is not from Cabrelli & Rothman (2010: 280), but from Cabrelli (2016/2017: 699), who in turn refers to info in a text passage in C&R, p. 280.

p.12, l. 533: Table 1. Should be Table 2.

p.17, l. 678: the groups’. Should it be the adult group’s? Compare lines 679-680.

p.20, l. 739: adolescent participants. Should be adolescent and adult participants.

Round 2

Reviewer 1 Report

The revised edition has improved in many ways. I appreciate the application of the GAM-analysis (even though there was not apparent difference with the LMER), which was carried out well as far as I can judge. The error bars may make the graphs a bit blurry, but provides essential information about variability, so as far as I'm concerned a very useful change. I understand that not all points could be addressed within the current context and time frame, but the paper is really nice in its revised form!

Author Response

Dear Reviewer, 

Again, thank you so much for your close reading of my manuscript and taking the time to provide such helpful feedback. 

Best wishes!